# The Influence of Forest Resting Environments on Stress Using Virtual Reality

**DOI:** 10.3390/ijerph16183263

**Published:** 2019-09-05

**Authors:** Xiaobo Wang, Yaxing Shi, Bo Zhang, Yencheng Chiang

**Affiliations:** 1College of Architecture and Arts, North China University of Technology, Beijing 100144, China; 2Beijing Laboratory of Urban and Rural Ecological Environment, Beijing Municipal Education Commission, Beijing Forestry University, Beijing 100083, China; 3Beijing Forest Well-being Planning and Design Research Co., Ltd., Beijing 100083, China; 4Department of Landscape Architecture, National Chiayi University, Chiayi City 60004, Taiwan

**Keywords:** forest, resting environment, stress, water, forest therapy

## Abstract

To explore the effects of different types of forest environments for forest therapy, this study focused on forest resting environments. Seven representative forest resting environments found in field research in Beijing were used as independent variables and were shown to subjects by a virtual reality (VR) video. Stress level was used as the dependent variable, and blood pressure, heart rate, salivary amylase, and the Brief Profile of Mood States (BPOMS) were used as physiological and psychological indicators. A between-subjects design was used in the experiment. A total of 96 subjects were randomly assigned to each environment type, and only one type of forest resting environment was observed. Through the relevant sample *t*-test and one-way analysis of variance, the pre- and post-test data of the indicators were analyzed. This study found that all the seven different types of forest resting environments can produce stress relief effects to some extent. Different types of forest resting environments have different effects on relieving stress. The most natural environment does not have the most significant effect on stress relief. A water landscape has a positive effect on the relief of stress. The conclusions of this study are conducive to the better use of the forest environment for forest therapy services.

## 1. Introduction

### 1.1. Forest, Stress, and Health

In recent years, more and more studies have proved that the natural environment can promote people’s physical and mental health [1,2,3,4]. Forests have been studied many times as typical natural environments [5,6,7]. These studies directly prove or infer that the forest environment can promote human health, perhaps through relieving stress [8].

Stress has been used multiple times as a health-related variable in many studies. Studies have shown that long-term stress can have a negative impact and seriously affect people’s health [9,10,11]. Many mental disorders and cardiovascular diseases are closely related to stress [12,13,14]. Ulrich proposed the theory of stress relief, from the perspective of evolutionary psychology, which states that exposure to the natural environment can reduce stress and have a direct effect on cognitive recovery [15]. Exposure to a real or simulated natural scene can restore a person’s psychology and physiology from stress by enhancing positive emotions and reducing negative emotions, such as fear or anger [16]. A study has shown that nature promotes health, mainly through four ways—air quality, physical activity, social cohesion, and stress reduction [4].

Many studies have shown that forests contribute to health. The forest environment has been shown to be effective in relieving stress and depression and in giving psychological relief [17,18,19,20]. Studies based on the physiological effects of relaxation in the forest environment have shown that rest in the forest can reduce cerebral blood flow in the pre-frontal cortex [5], lower blood pressure and heart rate [21,22], increase parasympathetic activity, inhibit sympathetic activity [14,15,20,21], reduce salivary cortisol stress hormone concentrations [21,22], and enhance the activity of natural killer cells and anticancer proteins [6,23] to improve immune function.

### 1.2. Research on Different Types of Natural Environments Related to Forests

The existing research studies on forests to promote people’s physical and mental health are mostly comparative studies of the forest environment and urban environment. Only a few studies have investigated the recovery effects of different green environments or characteristics [24,25]. A review [26] showed that research has involved high or low openness in the forest, with or without water in the forest, contrasting forests and golf courses, etc. Staats et al., studied the different density and accessibility of forest landscapes, finding that there was higher pleasure for higher accessibility and no significant difference related to density [27]. Through interviews and questionnaires, Herzog et al., conducted a comparative study of forests and cities with different degrees of openness. The study found that, in the forest, high openness made people feel calm and low openness made people feel afraid, while in cities, it was the opposite [28]. Van et al., studied park-like forest with and without a creek and found no difference between environments with and without water [29]. Other studies have investigated different qualities of forest environments, including the effects of different managed forest types [30], stand density [31], and vegetation types [8].

According to the different perceived sensory dimensions (PSDs) of environments, there have been a few studies with conflicting conclusions [24,32]. One study found that serene, rich in species, refuge, and nature PSDs were considered to have the most recovery value and that space was also important [33]. Another study of individuals with a stress-related mental disorder believed that social quietness was important and that the perceived sensory dimensions seemed to be of equal importance [34]. However, a new study in China suggested that the environment for stress recovery would be multilayered woodlands with water, serenity, nature, less prospect, little or no culture, and a social dimension [35].

Earlier research has shown that there are eight main characters of parks and gardens. They are serene, wild, rich in species, space, the common, the pleasure garden, festive, culture. And all the characters were included in the healing garden for people suffering from burnout diseases in Alnarp (Sweden) [36].

Some studies have focused on the impacts of different landscape types in relation to forests or woods. A study showed that urban parks, woodlands, and wild forests had more health benefits than streetscapes [37]. In one study, four different natural landscape types for stress relief were studied, and the results showed that exposure to the natural environment was much better than the built environment [38]. In another, urban parks and urban forests had a similar positive impact for stress reduction, but the urban forest seemed more positive [39]. These studies might suggest that the more natural the environment is, the better stress relieving effect it has. More natural means the presence of significant amounts of trees, shrubs, water, and other natural elements with minimal evidence of human influence [38].

### 1.3. Forest Therapy and Resting Environment

Forest therapy has emerged in the context of research on forest health benefits. It is called “shinrinyoku” in Japan and is developing rapidly in China. Forest therapy is a health promotion method that is based on the forest environment and activities such as walking and rest. Forest therapy has been proven to have a favorable influence on the health of the human body and mind in many medical studies [40,41,42,43,44,45,46,47,48,49].

Forest rest is one of the essential activities in forest therapy. Forest rest is an activity that involves taking a rest in a certain place in the forest, which is compared to a forest walk in a forest bath. Forest rest usually includes activities such as body relaxation, body scanning, and meditation. In a previous study, the forest rest included deep breathing, abdominal breathing, lie down, and rest [42]. Contrast experiments related to forest therapy, usually included viewing scenery in a seated position, which was another kind of forest rest [50]. Although forest rest is rich in variety, it has similar requirements for the environment, such as quietness and privacy. Forest therapy activities, such as forest walks and forest sports, which are in contrast to forest rest, also require significant differences in environmental conditions. Indeed, a multitude of activities, specifically meditation, affect the stress relief effect, but the present study only focused on the environment of forest rest in order to assess the health benefits of the environment itself.

The quality of the forest resting environment directly affects the effects of forest recuperation and the experience of visitors. The forest resting environment could be chosen from many environmental resources in the forest. However, not all natural environments have equal recovery effects. This study tried to discover the different effects of different forest resting environments.

### 1.4. Application of Virtual Reality (VR) in Health- and Environment-Related Research

The definition of virtual reality (VR) is based on the concepts of “presence” and “telepresence”, which refer to the sense of being in an environment generated by natural or mediated means [51]. It mainly includes a simulation environment, sensation and perception, natural skills, and sensing equipment. VR can be used in many fields, such as medical, education, arts, entertainment, and military [52].

VR is widely used in the field of health. VR has been used for motor rehabilitation [53], for the improvement of functional recovery poststroke [54], for hand rehabilitation after a stroke [55], in the treatment of autism [56], etc.

In the field of research related to the environment, scholars have used window views, photos, video/videotapes, photographic slides, on-site experiences, etc. as experimental materials [2,15,25,27,57,58,59,60]. Although there are famous, classic studies among these experiments, VR has its unique advantages. VR technology is a big improvement in research methods, since it gives a three-dimensional experience, which is more realistic than the two-dimensional picture or video. At the same time, VR has better operability than exposure in a real environment. VR has the advantages of being close to the real experience, reducing input manpower and material resources, and can better control the independent variables. Thus, VR has the potential to be used more and more in environment-related research [61].

### 1.5. Study Purpose and Hypothesis

This study focused on the impact of different types of forest resting environments on human stress levels. The study aimed to explore the health benefits of different forest resting environments from the perspective of stress relief. A total of 33 forest parks, almost all the municipal level of forest parks in Beijing were field surveyed in this study. Based on site surveying, seven typical different types of forest resting environments were found. These seven environmental types had different characteristics in the eight PSDs. Stress-related experimental studies were conducted for these different types of forest resting environments. The relief effects of different forest resting environments on stress were compared and analyzed.

At present, there are few studies on different types of forest environments. As far as we know, there have been no studies on different forest environments that specifically target forest rest.

Research on the impact of different forest environments on human health is crucial for transforming the forest environment, carrying out forest therapy activities, and giving the public a choice of recuperation sites. Forest rest requires the use and transformation of the environment to better utilize the resource advantages of the forest. The development of forest therapy activities also requires more in-depth theoretical support so that the forest can be better utilized. At the same time, the understanding of the health benefits of different forest environments is also important for the public to choose the right place for recuperation. A review study indicated that research on different types of natural environments is one of the future directions of health-related environmental research [4].

The hypotheses that guide the present research are as follows:(1)All the seven types of forest resting environments could contribute to the relief of stress, reflected both in the physiological and psychological index;(2)Different forest resting environments (structure, wood, wood with bench/platform, waterfall/pool with plants, etc.) have different effects on stress relief;(3)The more natural factors there are, the more obvious the stress relief effect;(4)A water landscape plays an important role in relieving stress.

## 2. Methods

### 2.1. Participants

Participants were recruited through online posting. The principles of enrollment were as follows: aged between 18 and 35 years old and physical and mental health. At the time of recruitment, the personal data of the participants were registered. According to the health questionnaire, 96 college students and social workers who met the experimental requirements were selected as effective participants, including 33 males and 63 females. The average age of all subjects was 24.03 ± 5.29 years old, and the dominant hand was the right hand. All the subjects had normal vision or corrected normal vision, no mental disorder, no stress disorder, no abnormal organic disease, no brain trauma, and no endocrine diseases. This study was approved by the ethics committee of the North China University of Technology (no. 51708003). All participants signed the experimental informed consent form before the experiment.

### 2.2. Materials

The experimental material of this study were the VR images of seven forest resting environments. Based on field research conducted in 33 forest parks in Beijing (including 15 at the national level and 18 at the municipal level), seven different types of forest resting environments were extracted. The landscape elements, location, characteristics, and corresponding PSDs are shown in Table 1 and Figure 1. VR videos were filmed from May to August 2017 by a UCVR EYE-01 camera (Pinkang Smart Company, Changzhou, China). The videos all had the same weather conditions, time, and shooting methods. The method of shooting was to set the bracket at a height of 1.4 m from the human point of view and fix the camera on the bracket in the same environment for 20 min at a 150° angle [62]. All videos were filmed in good weather conditions and showed no visitors or constructed facilities. Every video length was 5 min, in accordance with previous studies [57,61].

### 2.3. Measures

#### 2.3.1. Physiological Index

##### Blood Pressure and Heart Rate

Blood pressure and heart rate have usually been used as indexes to reflect people’s stress level [1,63,64], especially with regards to the difference in the data of the pre-test and the post-test. In this study, a HEM-7111 electronic sphygmomanometer (upper arm, OMRON, Dalian, China) was used to measure blood pressure and heart rate.

There were three tests of blood pressure and heart rate, reflecting normal conditions, the high stress state (after the Trier Social Stress Test), and the postintervention situation (after watching a certain type of forest resting environment). Blood pressure included the systolic blood pressure (SBP) and the diastolic blood pressure (DBP).

##### Salivary Amylase

Salivary amylase is an enzyme in saliva that can hydrolyze starch, which can reflect in stress levels. Since changes in the sympathetic nervous system can affect the secretion of salivary amylase, salivary amylase increases as the sympathetic system is activated. Related studies have found that salivary amylase is more rapid than hormones such as norepinephrine and cortisol and is a sensitive indicator of sympathetic nervous system excitability [65]. A study found that during the application of psychological stress, the salivary amylase of the subject increased [66].

This study used a professional saliva collection tube (salivette, SARSTEDT, Sarstedtstraße, Germany) to collect the test saliva. The subjects took the absorbent cotton strips in the collection tube and tipped them into their mouth for 1 min to collect the saliva. After chewing, the cotton strips were spat back into the tube. Then the test tube was stored in a freezer at −18 ℃. When all the experiments were finished, the saliva was analyzed by enzyme-linked immunosorbent assay (ELISA).

Two tests of salivary amylase were performed in the experiment. The first one was to reflect the high-pressure state after the Trier Social Stress Test (TSST). The second one was to reflect the pressure level after watching a certain type of forest resting environment.

#### 2.3.2. Psychological Index

##### Brief Profile of Mood States (BPOMS)

The Profile of Mood States (POMS) is an effective tool to research mood state which has a high degree of reliability and validity. It is widely used in clinical, pharmacodynamics, and psychology fields. The first version was compiled by McNair in 1971 with 65 items [67]. The Brief Profile of Mood States (BPOMS) was a simplification and a revision of POMS with 30 items [25]. BPOMS uses a five-point Likert scale format [26,27].

BPOMS includes five dimensions, namely tension (T), anxiety (A), fatigue (F), vigor (V), and confusion–depression (C + D). Total mood disturbance (TMD) is an important index of BPOMS (Equation (1)): TMD = T + A + F + C + D − V(1)

A high TMD score indicates a poor emotional state, there were two tests of BPOMS in this study, just like for the tests of salivary amylase. When the participants filled in the scale, they were asked to record their mood at that time. When they filled in the scale a second time (after watching a certain type of forest resting environment), they were asked to imagine that they were in the environment and could experience the real forest.

### 2.4. Procedure

This study used an independent group design—a between-subjects design. Participants were randomly assigned to seven different forest resting environments. The number of participants in each group is shown in Table 2. Each subject participated in one experiment alone to avoid interference with each other.

This study used the Trier Social Stress Test (TSST) to increase the stress level of the subjects. The TSST is a widely accepted laboratory psychosocial stress research paradigm [68]. In the present study, the TSST included a 5 min public speaking and 5 min public mental arithmetic task.

Participants were asked to pay attention to some items the day before the experiment, including eating breakfast; sleeping early; avoiding alcohol, tobacco, and rehabilitation drugs at least 24 h before the experiment; and avoiding intense activities or caffeine at least 12 h before the experiment. They were not to eat any food or drink liquid (except water) for at least 1 h before the experiment.

Before the experiment, there was a brief introduction, including the purpose of the experiment, the process and method, the risks and discomforts, and the confidentiality issues. During the experiment, the subjects were first measured for blood pressure and heart rate as the baseline, which enabled the subject to become familiar with the device. Then, the TSST test was conducted, and the participants were asked to perform a 5 min public speaking and a 5 min public mental arithmetic task, which were designed to increase the stress level of the participants. After the TSST, the pre-test was performed, including blood pressure, heart rate, salivary amylase, and the BPOMS scale. The pre-test of each indicator reflected the stress level at high pressure. After this, subjects were randomly assigned to watch a VR video of a forest resting environment for 5 min, a process that was assumed to relieve stress to some extent. Then, the post-test was carried out, in which the indicators were the same as the previous test. The indicator data for the post-test reflected the stress level after experiencing the intervention of a certain forest resting environment. The detailed process is shown in Figure 2.

During the experiment, the subjects wore second-generation VR glasses of the illusion mirror type to watch some kind of forest resting environment. While wearing the glasses, an adjustment of the distance between the eyes and the distance of the pupils was carried out according to individual characteristics. After confirming that the glasses had been debugged for the best appreciation state, the video would be timed for 5 min. During the viewing, subjects were asked to sit freely in the chair, rest in a comfortable position according to their own habits and imagine themselves in the environment they were watching.

### 2.5. Statistical Analysis

The experimental data were compiled and statistically analyzed using the software SPSS 17.0 (International Business Machines Corporation, Armonk, New York State, USA) and EXCEL 2010 (Microsoft Corporation, Redmond, Washington State, USA). First, the data were judged whether they conformed to the normal distribution through the Q-Q diagram, and then the homogeneity test of variance was performed. Then two main analytical methods were used: (1) for the pre- and post-test data of the same indicator, a relevant sample *t*-test was performed to determine whether the intervention of the forest resting environment played a role in relieving stress and (2) for the same indicator, the difference of the pre- and post-test was used as the amount of change, ΔD (ΔD = pre-test – post-test). The ΔD was analyzed by the next two methods to determine whether there were differences between different types of forest resting environments. For the condition that the variation satisfied the homogeneity of variance, a one-way ANOVA and a LSD (used for pairwise comparison) post-hoc multicomparison were performed for the ΔD. For the condition that the variation did not satisfy the homogeneity, a nonparametric test was performed.

## 3. Results

There were no missing data. We analyzed the outliers using a box plot. It was found that there were some outliers for the pre-test and post-test values of the indicators. However, through comparison observation, it was found that the pre-test and post-test had certain correspondence. We judged that these outliers were not errors in measurement, recording, or experimental design. In order to maintain the sample size and authenticity of the data, we chose to keep these outliers.

The data were analyzed, including the age, gender, baseline SBP/DBP, and heart rate. No significant differences were found between the types of individuals. From a statistical point of view, it might be considered that there was no difference between the individuals.

### 3.1. Physiological Index

Due to funding constraints, the basic measurements only collected data on blood pressure and heart rate. The baseline data were analyzed. We found that the DBP and heart rate values between the groups were in a normal distribution, satisfying the homogeneity of the variance, so the ANOVA test could be performed. The results were: DBP, F (6, 89) = 1.724, *p* > 0.05; Heart Rate, F (6, 89) = 0.543, *p* > 0.05. SBP did not satisfy the homogeneity of variance, so a nonparametric test was performed. The results of the Kruskal–Wallis test were *p* > 0.05. The above results indicated that there was no significant difference between the baseline between the groups, i.e., the physiological index of the subjects in each group could be considered to be consistent.

The baseline and the pre-test data were compared by a related sample *t*-test. The results were: SBP, t (95) = −2.242, *p* < 0.05; DBP, t (95) = −2.115, *p* < 0.05; Heart Rate (basic) − Heart Rate (pre-), t (95) = −2.077, *p* < 0.05. The results showed that the pre-measurement values of blood pressure and heart rate had increased relative to the baseline values. This showed that the TSST test had played a role in increasing stress.

#### 3.1.1. Comparison of Pre-Test and Post-Test

The data of the pre-test and post-test of each group were in accordance with the normal distribution by Q-Q diagram analysis, and the homogeneity of variance was satisfied. The relevant sample *t*-test could be performed.

Statistical differences were found in four types of forest resting environments between the pre-test and post-test data. They were: Type 3, DBP, t(11) = 3.36, *p* < 0.05; Type 4, DBP, t(13) = 2.84, *p* < 0.05 and salivary amylase concentration, t(13) = 3.51, *p* < 0.01; Type 5, SBP, t(11) = 2.21, *p* = 0.05 and DBP, t(11) = 2.51, *p* < 0.05; and Type 6, SBP, t(14) = 3.64, *p* < 0.01 and DBP, t(14) = 2.23, *p* < 0.05. The others did not have statistically significant differences.

Regarding trends, only the blood pressure and heart rate of Type 1 increased, while all the other types decreased. The post-test value of the salivary amylase concentration of Type 4 was significantly increased (Figure 3 and Figure 4).

#### 3.1.2. Comparison of ΔD

The study found that the overall variance of the difference between systolic and diastolic blood pressure was not uniform. A two-two group analysis of the data was performed. Nonparametric tests were performed on data that did not satisfy the homogeneity of the variance, and an ANOVA was performed on the data satisfying the homogeneity of the variance. The results showed that except for environments Type 1 and Type 6, the ΔD among the other environments was not significant. The data of the ΔD between Type 1 and Type 6 of the SBP and DBP satisfied the homogeneity of variance, and an ANOVA was performed: SBP F (1, 28) = 5.607, *p* < 0.05, DBP F (1, 28) = 4.854, *p* < 0.05.

The ΔD of SBP and DBP in different types of forest resting environments was as follows: Type 6 had the greatest impact, where the SBP and DBP decreased the most (Figure 5).

The ΔD of the heart rate satisfied the homogeneity of the variance, and ANOVA and the multiple comparisons of the LSD were performed. The significant differences were as follows: Type 1 and Type 3—*p* < 0.05, Type 1 and Type 6—*p* < 0.05, and Type 1 and Type 7—*p* < 0.05 (Table 3).

The heart rate mean value of the ΔD in different forest resting environments was as follows: Type 3 reduced the most, followed by Type 6 and Type 7, and Type 1 (the pure artificial landscaping environment) was the only type with an increased heart rate (Figure 6).

The ΔD of the salivary amylase satisfied the homogeneity of the variance, and ANOVA and multiple comparisons of the LSD were performed. The significant differences were as follows: Type 4 and Type 3—*p* < 0.05, Type 4 and Type 6—*p* < 0.05, and Type 4 and Type 7—*p* = 0.01 (Table 4).

The mean value of the ΔD of salivary amylase concentration in different forest resting environments was as follows: Type 4 was the largest and the change was increasing, which might indicate that the environment increased the stress, followed by Type 1 that also showed an upward trend. Furthermore, the biggest declining trend was found in Type 7 (Figure 7).

### 3.2. Psychological Index

#### 3.2.1. Comparison of Pre-Test and Post-Test

The TMD values of all types of forest resting environments were reduced. All but Type 3 reached a significant level, with Type 1 and Type 6 reaching a significant level of 0.01 (Figure 8).

From the changes of the five dimensions of the pre-test and post-test of the BPOMS, the dimension values of Type 1 were significantly different; the dimension values of T and C + D of Type 2, Type 4, and Type 5 were significantly different; the V dimension of Type 3 was significantly different; except for the A dimension value, the other dimension values reached a 0.01 significant level; and the A and C + D dimension values were significantly different.

Regarding the trend of the pre-test and postmeasurement values, except for the V dimension value, the dimension values reduced. For the V values, Type 1, Type 3, and Type 6 increased significantly, while Type 4, Type 5, and Type 7 decreased (Figure 9).

#### 3.2.2. Comparison of ΔD

The ΔD of the TMD and every dimension of the BPOMS were analyzed by Q-Q diagram, which conformed to the normal distribution and satisfied the homogeneity of variance. ANOVA and multiple comparisons of LSD after the event of the ΔD found that: for TMD, Types 1 and 3 had significant differences; for A, Types 3 and 7 had significant differences; for V, Types 1 and 4/7, Types 3 and 4/5/7, and Types 6 and 4/5/7 had significant differences. This was important for discovering the rules and developing the discussion of different types. Taking TMD as an example, the LSD multiple comparison found that there was a significant difference between Type 1 and Type 3, *p* < 0.05.

For the mean value of the ΔD, different forest resting environments had different effects on TMD and different dimensions. For TMD, Type 1 and Type 6 had the greatest impact; for A, Type 7 had the greatest impact and Type 3 had the smallest; and for V, Type 6 had the greatest impact, followed by Type 3, and then Type 1, while Types 4, 5, and 7 decreased, with the trend of Type 7 being the most obvious (Figure 10).

## 4. Discussion

### 4.1. All the Seven Types of Forest Resting Environments Could Relieve Stress to Some Extent

The correlation sample *t*-test on the pre-test and post-test values of each indicator showed that all the seven types of forest resting environments could have certain effects on relieving stress. For Type 1 (structure), all the indicators of the BPOMS were significant and showed a positive effect on the improvement of mood. For Type 2 (wood), significant relief effects were found in two dimensions (tension and confusion–depression) of the BPOMS. For Type 3 (wood with bench), the diastolic blood pressure decreased significantly, and the vigor dimension of the BPOMS increased significantly. For Type 4 (wood with platform and bench), although the concentration of the salivary amylase increased, the diastolic blood pressure, tension, and confusion–depression decreased significantly. For Type 5 (platform with trees), the diastolic blood pressure, systolic blood pressure, tension, and confusion–depression decreased significantly. For Type 6 (waterfall with trees), the diastolic blood pressure and systolic blood pressure decreased significantly and the BPOMS index reached a significant level of 0.01, except for the anxiety dimension. For Type 7 (pool with plants), the anxiety and confusion–depression dimensions decreased significantly.

These results might be due to the fact that there were many plants and other natural factors in the VR videos of the seven types of forest resting environments. This is consistent with the conclusions of a large number of studies related to the natural environment, including stress reduction, adjustment of emotional state, the positive relationship between the natural environment and health parameters, the positive effect of the forest on central nervous system activity (including autonomic activity), endocrine activity, and immune function [2,15,69,70,71].

### 4.2. Different Forest Resting Environments have Different Effects on Stress Relief

The ANOVA or nonparametric tests of the ΔD showed that different forest resting environments had different effects on stress relief:On physiological indicators:

For the diastolic and systolic blood pressures, the comparison of the mean of the ΔD revealed that there was a significant difference between Type 1 (structure) and Type 6 (waterfall with trees).

For the heart rate, the analysis of the ΔD showed that there was a significant difference between Type 1 (structure) and Types 3 (wood with bench)/6 (waterfall with trees)/7 (pool with plants).

For the salivary amylase, the analysis of the ΔD showed that there was a significant difference between Type 4 (wood with platform and bench) and Types 3 (wood with bench)/6 (waterfall with trees)/7 (pool with plants).

In summary, the stress relief effects of Type 1 (structure) and Type 4 (wood with platform and bench) were very different from those of Type 3 (wood with bench), Type 6 (waterfall with trees), and Type 7 (pool with plants). The former were far less effective than the latter.

Among the seven different types of forest environments, Type 1 was dominated by buildings and structures and Type 4 had podiums/wood platforms and benches. These two types of environments had many artificial features, especially Type 4 with its performance and podium functions, which might lead to tension for Chinese subjects. Type 3 was a forest and had a more natural wooden bench environment. Type 6 was a plant, rock, and waterfall environment. Type 7 was a plant, stone barge, and pool environment. These three types of environments were more natural than Types 1 and 4.

These results were consistent with previous research that, for stress relief, the forest environment is better than the urban environment [21,40,63], which indicates that the natural environment is more active than the artificial environment.
On psychological indicators:

For TMD, the comparison of the mean of the ΔD showed that there was a significant difference between Type 1 (structure) and Types 3 (wood with bench)/4 (wood with platform and bench).

For the anxiety dimension of the BPOMS, the difference between Type 3 (wood with bench) and Type 7 (pool with plants) was significant.

For the vigor dimension, there were significant differences between Types 1/3/6 and Types 4/5/7 and between Type 2 and Types 4/7.

Type 1 (wooden pavilion, building, wooden platform, gravel paving, plants, tables and chairs, etc.), which had the highest degree of artificialization, decreased the total mood disturbance much more than Type 3 (forest with wooden chairs) and Type 4 (forest with podiums/wood platforms, and benches).

Type 7 with still water decreased the anxiety dimension much more than Type 3.

In terms of the vigor dimension, Types 1, 3, and 6, which played a similar role, caused a significant increase. This was in stark contrast to Types 4, 5, and 7, which caused a reduction.

### 4.3. Not the Most Natural Forest Resting Environment Gives the Most Effective Stress Relief

It is worth noting that among the seven types of forest resting environments, the most natural environment was Type 2 (wood) without any artificial facilities—only plants, land, and sky. However, the results showed that, for both physiology and psychology, Type 2 was not outstanding. In addition, it only had a significant effect on the tension and confusion–depression dimensions of the BPOMS.

For physiology, Type 3 (wood with bench), Type 5 (platform with trees), and Type 6 (waterfall with trees) showed significance. These three types of environments all had some manual intervention. There were wooden benches in Type 3 and a large wooden platform in Type 5. Type 6 was relatively natural, but it was artificially natural with artificially planted plants, stacked rockeries, created waterfalls, etc.

For psychology, Type 1 (structure) and Type 6 (waterfall with trees) were outstanding. Indeed, Type 1 was the most artificial environment, full of buildings and structures. However, Type 1 had a very significant effect on the regulation of emotions.

These results are in contrast to some of the conclusions that stress relief in the natural environment is best [38,39,60,72]. However, some studies have suggested that it is not the case that the more natural the environment is, the better the recovery effect. Historical blocks, monasteries, art galleries, shopping centers, and coffee houses might all have a recovery effect [73,74,75,76].

According to Kaplan’s research, the environment for better recovery is rich in content [17,77]. It is speculated that the environment with rich content might have a better effect on relieving stress. For Type 3, besides the forest, there are wooden seats to enrich the content of the pure natural environment. For Type 1, the content elements are abundant and refuge characteristics could be found. Perhaps these are the reasons that explain the results.

### 4.4. Water Landscape Has a Positive Effect on Stress Relief

The water landscape has a positive effect on the relief of stress, but this effect cannot be generalized. Among the seven environments, Type 6 (waterfall with trees) contained moving water and Type 7 (pool with plants) was mainly composed of still water. Type 6 was the most prominent type of environment for stress relief among all environments, both physically and psychologically. For Type 6, physiologically, it caused a significant decrease in diastolic and systolic blood pressure and, psychologically, the effect of relieving psychological stress was obvious—all except the anxiety dimension reached a significant level of 0.01. For Type 7, physiologically, although they did not reach a significant level, blood pressure, heart rate, and salivary amylase concentrations all showed a downward trend, with the salivary amylase concentration declining the most among the seven types of environments. Psychologically, Type 7 was somewhat different from Type 6 and caused a significant drop in the dimensions of anxiety and confusion–depression; additionally, the vigor dimension decreased but did not reach a significant level.

These results are evidenced by the conclusions of some previous studies. For physical and psychological health, a “blue” gradient has been found. In self-perception, the closer to the ocean, the more the effect of water exceeds the green environment effect [78,79]. Compared to exercising in urban green spaces, on farmland, and in woodland environments, exercise in the environment near water has been shown to better improve the mood [80]. Water environments, such as beaches and rivers, have been shown to produce better mental health and low negative emotions [81]. There has also been a study inconsistent with the conclusions of the present study. It found that there was no difference in happiness, stress, anger, depression or tension, with or without water, in a park-like forest environment [29].

### 4.5. Limitations of This Study and Future Research Direction

This study has some limitations, and should be improved in future research:

First, this study only explored the stress relief effects of seven types of forest resting environments. These seven types were only a summary of the survey of forest parks in Beijing. In an actual situation, the types of environments that could be used for forest therapy may be more diverse and should be expanded much more in future research.

Second, this study found that different types of forest resting environments did have different stress relieving effects, but the essential reasons for the differences were not found. That is, the internal mechanism responsible for producing the different stress relieving effects was not established.

The reasons for the stress relief effect may be naturalness, comfort, beauty, familiarity, peace of mind, spatial characteristics, etc. An in-depth study of these reasons will make the theory of environmental health promotion clearer and more conducive to guiding actual planning and design, which should be the focus of future research.

Third, the theoretical background for selecting the seven environments was not sufficient. Although we quoted PSD theory, the theory was not very compatible with the seven environments.

This study analyzed the characteristics of the seven environments in various PSDs. Type 1 (structure) had “refuge, social, and space” characteristics, Types 2 (wood)/3 (wood with bench)/6 (waterfall with trees) were “rich in species and nature”, Type 4 (wood with platform and bench) had “social and culture” characteristics, Type 5 (platform with tree) had “refuge”, and Type 7 (pool with plants) was “serene and prospect”. In this study, with regards to the stress relief effects of various environments, the pre-test and post-test differences of the physiological indicators of Types 2 (wood), 3 (wood with bench), and 5 (platform with trees) were significant. In addition, the pre-test and post-test differences of the psychological indicators of Types 1 (structure) and 6 (waterfall with trees) were significant. The stress relief effects did not establish a direct connection with the PSDs. However, the results of the study suggested that “refuge” (as Type 1 had) might have an important role in the psychological relief of stress.

Future research should pay more attention to the background of the theoretical framework, and the independent variables should be better designed and based on certain theories:

Fourth, the water landscape in this study showed some complexity for the relief of stress. The stress relief effect of water seems to be affected by many factors. Different types of waterscapes have different effects on relieving stress, which is worthy of further exploration in future research.

Fifth, the indicators need to be further improved. In this study, salivary amylase concentration increased to a certain extent, which was inconsistent with previous studies. This might be related to factors such as the small number of subjects and the distortion of watching VR video. In the future, the sensitivity index of heart rate variability (HRV) should be added, which will make the research results more objective.

Sixth, the visual angle of the VR video was fixed, so aesthetic fatigue might occur in the later stages of viewing. At the same time, wearing VR glasses was not the same as the real environment experience. In the future, field experiments should be implemented as much as possible to increase the ecological validity of the study.

Seventh, a forest environment is a place full of the five senses—hearing, smell, taste, touch, and sight. Additionally, air (especially phytoncide and negative oxygen ions), temperature, humidity, etc. could also have an effect on stress in the real field. However, this study only focused on visual factors. Other factors should be incorporated in future research.

Eighth, the specific activities for “forest rests” were ignored in this study. There are different kinds of activities for forest rests, such as meditation and body scanning, and more activities for forest therapy, such as walking and painting. These activities could also affect stress relief. Studies focused on the health benefit of different kinds of activities of forest therapy are critically needed.

Ninth, most of the participants were undergraduate students and could not be representative of the general public who participate in forest therapy. The number of participants also needs to be increased.

## 5. Conclusions

The environments for resting in the forest shown by VR video can produce stress relief effects to some extent;Different types of forest resting environments have different effects on relieving stress. The artificial environment is not as effective as the natural environment in relieving stress physiologically, however, the artificial environment can bring about recovery psychologically;The forest resting environment that gives the best stress relief is not the most natural. With a certain amount of manual intervention, the natural environment containing some facilities has the effect of relieving stress better than the pure natural environment;A water landscape has a positive effect on the relief of stress, especially a dynamic water landscape, which can play a good role in relieving stress both physically and psychologically.

## Figures and Tables

**Figure 1 ijerph-16-03263-f001:**
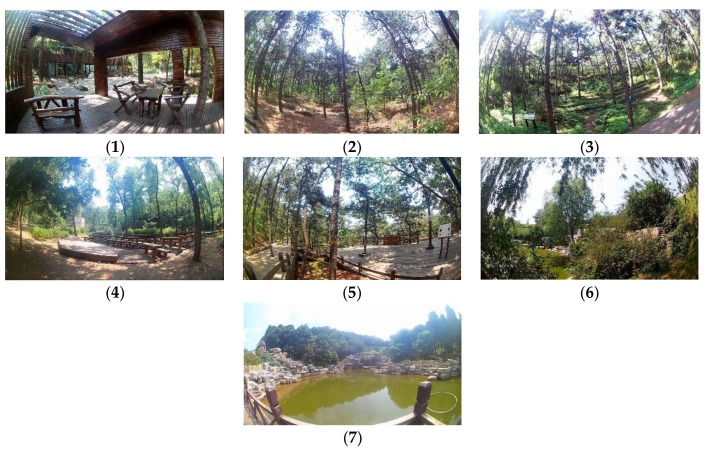
The videos of seven different types of forest resting environments: (**1**) structure, (**2**) wood, (**3**) wood with bench, (**4**) wood with platform and bench, (**5**) platform with trees, (**6**) waterfall with trees, and (**7**) pool with plants.

**Figure 2 ijerph-16-03263-f002:**
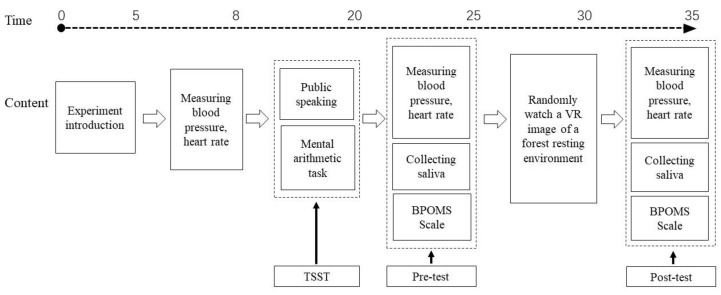
Experimental process chart. VR-virtual reality; TSST-Trier Social Stress Test; BPOMS-Brief Profile of Mood States.

**Figure 3 ijerph-16-03263-f003:**
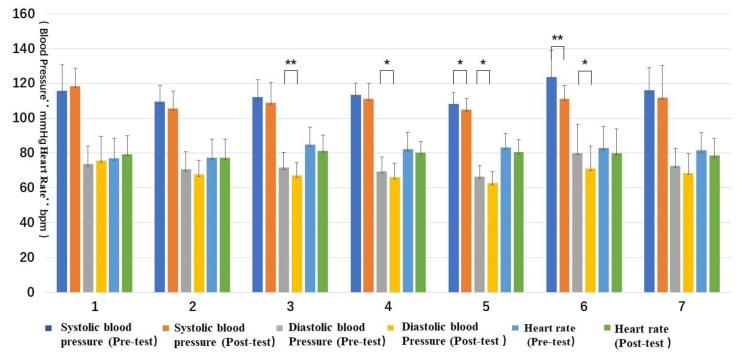
Pre- and post-test of blood pressure and heart rate: (**1**) structure, (**2**) wood, (**3**) wood with bench, (**4**) wood with platform and bench, (**5**) platform with trees, (**6**) waterfall with trees, and (**7**) pool with plants. Interpretation: * means *p* < 0.05, ** means *p* < 0.01.

**Figure 4 ijerph-16-03263-f004:**
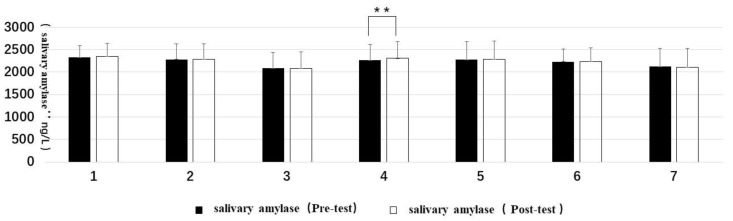
Pre- and post-test of salivary amylase: (**1**) structure, (**2**) wood, (**3**) wood with bench, (**4**) wood with platform and bench, (**5**) platform with trees, (**6**) waterfall with trees, and (**7**) pool with plants. Interpretation: ** means *p* < 0.01.

**Figure 5 ijerph-16-03263-f005:**
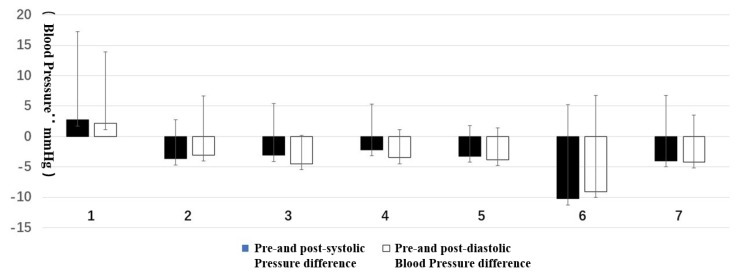
The mean of the amount of change (ΔD) of systolic blood pressure (SBP) and diastolic blood pressure (DBP): (**1**) structure, (**2**) wood, (**3**) wood with bench, (**4**) wood with platform and bench, (**5**) platform with trees, (**6**) waterfall with trees, and (**7**) pool with plants.

**Figure 6 ijerph-16-03263-f006:**
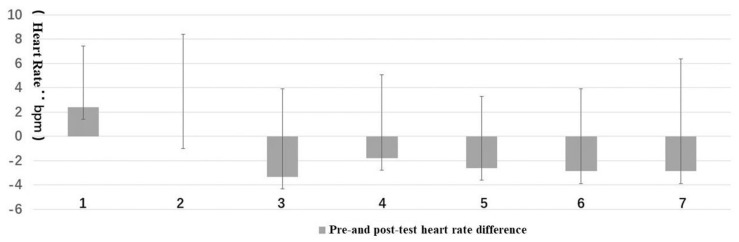
The mean of the ΔD of heart rate: (**1**) structure, (**2**) wood, (**3**) wood with bench, (**4**) wood with platform and bench, (**5**) platform with trees, (**6**) waterfall with trees, and (**7**) pool with plants.

**Figure 7 ijerph-16-03263-f007:**
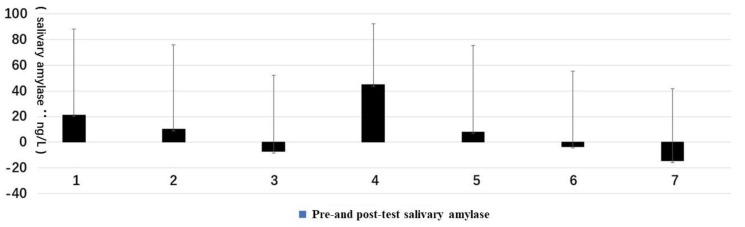
The mean of the ΔD of salivary amylase: (**1**) structure, (**2**) wood, (**3**) wood with bench, (**4**) wood with platform and bench, (**5**) platform with trees, (**6**) waterfall with trees, and (**7**) pool with plants.

**Figure 8 ijerph-16-03263-f008:**
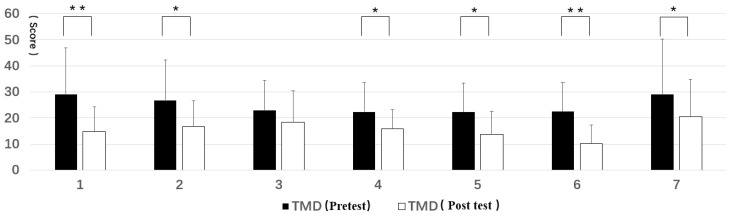
Pre- and post-test of total mood disturbance (TMD): (**1**) structure, (**2**) wood, (**3**) wood with bench, (**4**) wood with platform and bench, (**5**) platform with trees, (**6**) waterfall with trees, and (**7**) pool with plants. Interpretation: * means *p* < 0.05, ** means *p* < 0.01.

**Figure 9 ijerph-16-03263-f009:**
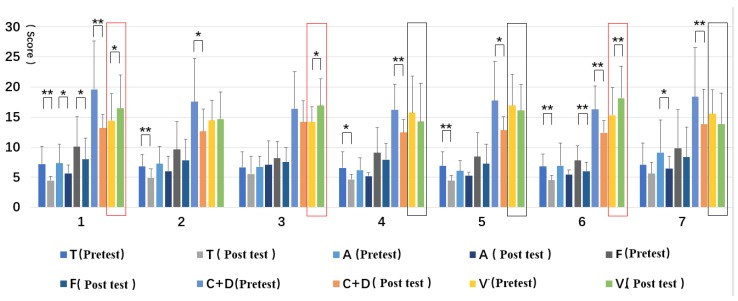
Pre- and post-test of each dimension of the BPOMS: (**1**) structure, (**2**) wood, (**3**) wood with bench, (**4**) wood with platform and bench, (**5**) platform with trees, (**6**) waterfall with trees, and (**7**) pool with plants. Interpretation: * means *p* < 0.05, ** means *p* < 0.01.

**Figure 10 ijerph-16-03263-f010:**
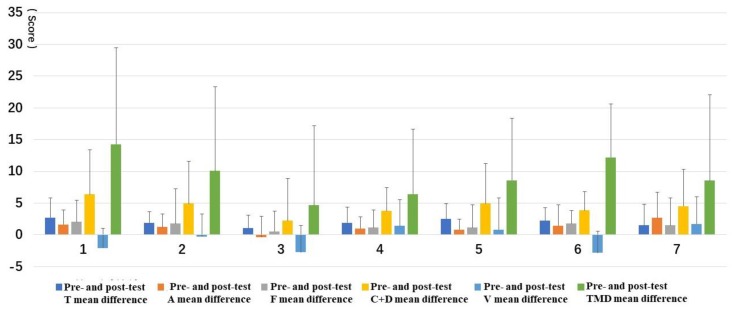
The mean value of the ΔD of the BPOMS: (**1**) structure, (**2**) wood, (**3**) wood with bench, (**4**) wood with platform and bench, (**5**) platform with trees, (**6**) waterfall with trees, and (**7**) pool with plants. T—tension, A—anxiety, F2014fatigue, V—vigor, C + D—confusion-depression.

**Table 1 ijerph-16-03263-t001:** Seven different types of forest resting environments.

Environment Type	Landscape Elements	Location	Environment Features	PSDs ^1^
1 (Structure)	Wooden pavilion, building, wooden platform, gravel paving, plants, tables and chairs	Outside the Forest Experience Hall of Beijing Badaling Forest Park	The highest degree of artificialization, waterless	Refuge, Social, Space
2 (Wood)	Plants, soil	Beijing Xishan Forest Park Forest Bath	The highest degree of naturalization, forest landscape, waterless	Rich in species, Nature
3 (Wood with bench)	Plants, wooden benches close to nature	Beijing Badaling Forest Park Forest Class	Similar to Type 2, forest landscape, more wooden benches, waterless	Rich in species, Nature
4 (Wood with platform and bench)	Plants, wooden platforms, wooden tables and benches, wood paving	Beijing Baiwangshan Forest Park Forest Bath	Space for performance or podium, waterless	Social, Culture
5 (Platform with trees)	Plants, wooden platforms, wooden planks, trees	Beijing Xishan Forest Park Forest Bath	Independent of the rest space of the trail, waterless	Refuge
6 (Waterfall with trees)	Plants, pool water, waterfall, rocks	Beijing Xishan Forest Park entrance waterfall	Plants in the near-middle prospect, hydrostatic and hydrodynamic	Rich in species, Nature
7 (Pool with plants)	Pool water, rocks, railing, fountain head, plants	Beijing Xishan Forest Park Pool	Distant view has plants, hydrostatic	Serene, Prospect

^1^ Perceived Sensory Dimensions.

**Table 2 ijerph-16-03263-t002:** Number of participants assigned to different forest resting environments.

Environment Type	1	2	3	4	5	6	7	Total
Number of participants	15	13	12	14	12	15	15	96

**Table 3 ijerph-16-03263-t003:** Post-hoc multiple comparison of LSD with significant differences in the heart rate of the ΔD.

Multiple Comparisons							
LSD							
Dependent Variable			Mean Difference (I–J)	Standard Error	Sig.	95% Confidence Interval	
						Lower Bound	Upper Bound
ΔD of heart rate	1	3	−5.73 *	2.79	0.04	−11.28	−0.18
		6	−5.27 *	2.63	0.05	−10.50	−0.03
		7	−5.27 *	2.63	0.05	−10.50	−0.03

* The mean difference is significant at the 0.05 level.

**Table 4 ijerph-16-03263-t004:** Post-hoc multiple comparison of LSD with significant differences in the salivary amylase of the ΔD.

Multiple Comparisons							
LSD							
Dependent Variable			Mean Difference (I–J)	Standard Error	Sig.	95% Confidence Interval	
						Lower Bound	Upper Bound
Pre- and post-test salivary amylase differences	4	3	−52.11 *	23.85	0.03	−99.51	−4.71
		6	−48.22 *	22.53	0.03	−92.99	−3.45
		7	−59.37 *	22.53	0.01	−104.14	−14.60

* The mean difference is significant at the 0.05 level.

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
