# Peer review of "The Influence of Forest Resting Environments on Stress Using Virtual Reality"

_ijerph, 2019, doi:10.3390/ijerph16183263_

Round 1
Reviewer 1 Report
Title: The impact of different forest resting environments on stress in Beijing
2019.7.25.
Comments to the Author
This is a curious study of comparing forest environments. But many parts must be improved with more explanations and discussions.
Major comments:
1. Forest environment is a place using the five senses such as sense of smell or hearing as well as sight, and air (including the phytoncide) and the calm temperature should be included as a factor. This study compares only using the sight in VR, and actually it isn't "a comparison of the different forest environment". If anything, the author should say that this study is "the comparison of the reaction by watching scenery of the different forest in VR".
2 . The author hypothesize that " the less the artificial factors, the more obvious the stress relief effect", but watching VR in a room is already a complete artificial state and is far apart from a natural state.
3 .The wavelength of natural sun light and the green color may be changed into different wavelength in VR. How does the difference influence on a result?
4 .In this study, an independent group design was adopted. But the result should be offset individual differences, so it may be better to compare the reaction of the same subject received plural sight stimulation.
Minor comment:
5. P5.line163 'The Profile of Mood Stares( POMS)→States
Author Response
Many thanks for the comments. Below are our point-by-point responses (in blue) to the comments. The page and line numbers please refer to our revised manuscript(ijerph-561044-R1.doc)
Major comments:
Forest environment is a place using the five senses such as sense of smell or hearing as well as sight, and air (including the phytoncide) and the calm temperature should be included as a factor. This study compares only using the sight in VR, and actually it isn't "a comparison of the different forest environment". If anything, the author should say that this study is "the comparison of the reaction by watching scenery of the different forest in VR".Response: This comment is so important to us, that we think we should change the title. The five senses and the air etc. of the forest environment are indeed very important for forest therapy. In order to control the independent variables, our study only focused on visual factors. We changed the title to “The influence of forest resting environments on stress using virtual reality”. Please see L3-4.
We also added the content about the five senses and the air etc. of forest environment in the discussion part, please see L507-510.
2 . The author hypothesize that " the less the artificial factors, the more obvious the stress relief effect", but watching VR in a room is already a complete artificial state and is far apart from a natural state.
Response: Yes, this is a big limitation of the study. VR in a room is so different from the real field. We discuss this in L503-506. However, due to financial, human and material constraints, VR is the closest experiment we can implement to the real scene.
Well, playing natural scenes through VR, if it is really effective for stress relief, maybe we could promote this VR natural product in the application field. Put the natural scenery into the VR, for example, we can put the forest into the VR, then they can be integrated into our daily life and play a natural health promoting effect. As far as we know, Japan once loaded the scenery of the forest into VR for people to experience. From this perspective, perhaps the application of artificial technology such as VR could be a promising means of bringing people and nature closer.
3 .The wavelength of natural sun light and the green color may be changed into different wavelength in VR. How does the difference influence on a result?
Response: The wavelength of natural sun light and the green color may indeed changed in VR. But we are concerned with the comparison of the differences between the three environments for stress relief. We balance the variables of the wavelength, that is, control the unrelated variables. The video shots were selected in the same weather conditions, time, shooting methods, sun angle, plant crown, height and so on. These methods are all trying to avoid wavelength changes that may lead to the differences influence on results.
4 .In this study, an independent group design was adopted. But the result should be offset individual differences, so it may be better to compare the reaction of the same subject received plural sight stimulation.
Response: The same subject received plural sight stimulation are indeed the best choice to counteract individual differences. However, for the present study, one type of forest environment experiment takes about 35 minutes. If all the three types do together, it may take too long times. And practice effect, fatigue effect, etc. would be emerged. That’s why we chosen the independent group design.
Well, thanks for the comment, we further analyzed the basic data, including the age, gender, basic SBP, DBP and heart rate. No significant differences were found between the types. From a statistical point of view, it might be considered that there is no difference between the individuals. This was supplemented in the manuscript, please see L256-272.
Minor comment:
P5.line163 'The Profile of Mood Stares( POMS)→StatesResponse: Corrected, please see L197.
Thank you so much for your valuable comments, which make our manuscript more rigorous and standardized. We feel that the manuscript has been greatly improved after revision based on your comments.

Reviewer 2 Report
I applaud the authors on collecting biomarker data on stress and comparing forest environments. This is an interesting and potentially important study; I enjoyed reviewing it. Comments are below.
L80 Citations and more discussion of specific activities for "forest rests" would provide important context for the study. Also, given the multitude of interventions here (i.e., meditation) how can the effects be attributed to forests alone?
L99 "construction of forest sanitation bases" is unclear to this North America reviewer and likely future readers
L107: "could contribute to the relief of stress" is not sufficiently specific for formal hypothesis testing, consider making these objectives. Also, review of different types of forests shold be incldued in the lit review, even though the authors say there isn't sufficient literature on it.
L109: "less the artificia factors, the more obvious the stress relief" is qute unclear. Artificial is needed to be discussed in the lit reivew.
L124: What is a "VR image"? There should be a larger section on VR in the lit review.
L132: Why was 5 min exposure chosen?
Fig 1: The selection of these quite different environments should be defended theoretically or empirically.
L177: The number of subjects per treatment is relatively low. A power analysis a priori is warranted. Should have considered a between subjects design which provides more statistical power with low sample sizes.
L198: Experimental process seems reaonsable and is similar to past studies looking at stress recovery from exposure to nature However, when was saliva taken? Cortisol takes 20 min to get into saliva, what about amalyse?
L226: This topic sentence is quite unclear. Also, the p-value reporting does not match standard convention. Where multi-variate outliers tested for? What about missing data? Why not compare for differences between change scores, rather than pre-post? And how are these physio measures really tested pre/post? That is more appropriate for survey data. That setting #4 was significant for saliva seems odd, since it doesn't have water and is very similar to #5. I question the analytical approach here, and the sample size per treatment.
In general, figures are quite clear. Consider descriptively labeling the images rather than using numbers (1-7) however, so readers don't have to keep scrolling up to see the pictures.
THorughout - shouldn't there be comparisons to baseline measures, not just pre/post exposure? Also, where are tests for differences in baseline measures? What about tests for the effect of the stressor, to ensure it elicited the desired effect?
Fig 10: Would be helpful to walk reader through this diagram more, and explain further why this approach is warranted.
L304: Again, descriptive labels would be helpful rather than numbers. And again, some theoretical justification for the selection of these is warranted to situate the study within the broader env psychology background.
L356: Some of the writing here is awkward.
L433: A discussion of the ecological validity of "VR" forest bathing is warranted as well.
L433: Awkward to have bullets for a conclusion...
Author Response
We appreciate your affirmation and encouragement very much, and we are very grateful for your valuable comments. We will take each item seriously and modify it one by one. The page and line numbers please refer to our revised manuscript(ijerph-561044-R1.doc).
I applaud the authors on collecting biomarker data on stress and comparing forest environments. This is an interesting and potentially important study; I enjoyed reviewing it. Comments are below.
L80 Citations and more discussion of specific activities for "forest rests" would provide important context for the study. Also, given the multitude of interventions here (i.e., meditation) how can the effects be attributed to forests alone?
Response: We provided more context of forest rests in the introduction part, and added discussions of specific activities for “forest rests”, please see L89-92, and L511-515. We very agree that the activities can affect the health benefits. Perhaps we might design a research specifically on activities of forest therapy in the future.
The choice of forest resting environment as a research object is because forest rest is an indispensable activity in forest therapy. Although it is rich in variety, it has similar requirements for the environment, such as quietness and privacy. Forest therapy activities, such as forest walks, forest sports, etc., which are in contrast to forest rest, require significant differences in environmental conditions. That’s the original intention of this study to choose the forest resting environment. Actually, the study of forest waling environment is one of the subjects what we have studied now.
L99 "construction of forest sanitation bases" is unclear to this North America reviewer and likely future readers
Response: We revised this, please see L128.
L107: "could contribute to the relief of stress" is not sufficiently specific for formal hypothesis testing, consider making these objectives. Also, review of different types of forests should be included in the lit review, even though the authors say there isn't sufficient literature on it.
Response: We revised the hypothesis, and tried to make it more objective. Please see L136-137. For the second hypothesis, it is a little difficult to descript different types of forests, because there are 7 types of this study. But we tried to make it clearer. Please see L138-139. And we think the introduction of study about PSDs and woodlands with water are important literatures about different types of forests. So, we put them a separate paragraph. please see L68-74. Also, we added some studies about the landscape types related to forest or wood. In order to make this part of the content more substantial. Please see L75-81.
L109: "less the artificial factors, the more obvious the stress relief" is quite unclear. Artificial is needed to be discussed in the lit review.
Response: We modified the expression of this sentence, and added some literatures in the introduction part. Please see L140 and L75-81.
L124: What is a "VR image"? There should be a larger section on VR in the lit review.
Response: We added a section of VR in the lit review. Please see L98-114.
L132: Why was 5 min exposure chosen?
Response: We according the previous study of Hartig, T., 1997 and Bing J et, al. 2014. Please see L163-164, L655-656, and L663-664.
Fig 1: The selection of these quite different environments should be defended theoretically or empirically.
Response: These different environments were given more description, both below the figure 1 and in the table 1.
L177: The number of subjects per treatment is relatively low. A power analysis a priori is warranted. Should have considered a between subjects design which provides more statistical power with low sample sizes.
Response: Yes, this study is a between subjects design, which we called an independent group design. They are two names of the same thing. We added some words to make it clearer. Please see L210
L198: Experimental process seems reaonsable and is similar to past studies looking at stress recovery from exposure to nature However, when was saliva taken? Cortisol takes 20 min to get into saliva, what about amalyse?
Response: Salivary amylase reacts much faster than salivary cortisol. The salivary amylase sample can be sampled in just 1 minute. Studies have found that salivary amylase begins to increase after the second minute of the TSST test; salivary amylase can slowly decrease after 3 minutes of exposure to a relaxed environment.
L226: This topic sentence is quite unclear. Also, the p-value reporting does not match standard convention. Where multi-variate outliers tested for? What about missing data? Why not compare for differences between change scores, rather than pre-post? And how are these physio measures really tested pre/post? That is more appropriate for survey data. That setting #4 was significant for saliva seems odd, since it doesn't have water and is very similar to #5. I question the analytical approach here, and the sample size per treatment.
Response: We reorganized the topic sentence to make it clearer, please see L277. We changed the reporting method of the p-value, please see L279-281.The outliers were eliminated by the box plot, and there was no missing data. The data analyzed in this manuscript were all valid values. We reported the comparation for differences between change scores as △D. For the pre- and post-test data of the same indicator, a relevant sample T test was performed to determine whether the intervention of the forest resting environment played a role in relieving stress. The differences between change scores were analyzed to determine whether there was difference between the types, and we put it in the next part after the pre- and post-test, please see L293-327, L346-367.
Setting #4 was a space for performance or podium, in which the platform was the focal point of vision that could be seen by anyone sitting on a bench. #5 was a rest space without performance and podium functions. We explain this in table 1, please see L165(4,5). Perhaps performance or podium could lead to tension, especially for the Chinese persons. You know most of us are shy in public. We added this in the discussion part, please see L405-406. Well #5 did not have such problem, on the contrary, the rest in a big platform might bring a good relaxation.
In general, figures are quite clear. Consider descriptively labeling the images rather than using numbers (1-7) however, so readers don't have to keep scrolling up to see the pictures.
Response: This is an important suggestion. We revised the description in the introduction and discussion section. Well, in aimed to make the results looked clear, we kept the numbers in the result section, and only added the description below the figures.
THorughout - shouldn't there be comparisons to baseline measures, not just pre/post exposure? Also, where are tests for differences in baseline measures? What about tests for the effect of the stressor, to ensure it elicited the desired effect?
Response: We did the basic measures in the experiment but did not report them in the manuscript. The revised manuscript supplemented these contents. Please see L256-272. TSST test is used to induce pressure. We compared the basic measurements (basic-test) with the data after the TSST (the pre-test of blood pressure, heart rate) and found a significant increase, indicating that the TSST did have an effect of increasing stress.
Fig 10: Would be helpful to walk reader through this diagram more, and explain further why this approach is warranted.
Response: We gave more explain about the figure, please see L351-355.
L304: Again, descriptive labels would be helpful rather than numbers. And again, some theoretical justification for the selection of these is warranted to situate the study within the broader env psychology background.
Response: We revised the description as the above showed.
For the theoretical justification, that’s our difficulty of this study. We found the theory of perceived sensory dimensions (PSDs). And tried to found some relationships with the types. But we didn’t observe obvious relationships. We described this in the introduction, materials and discussion sections. Please see L68-74, L165, L480-487. We think we need to read more, and find more theoretical justification in the future research. Thank you so much for pointing out such a critical issue, which gives us important insights.
L356: Some of the writing here is awkward.
Response: We revised the sentence, please see L429-430.
L433: A discussion of the ecological validity of "VR" forest bathing is warranted as well.
Response: Yes,that’s important. Please see L503-510.
L433: Awkward to have bullets for a conclusion...
Response: We revised it, please see L520.
In sum, we sincerely appreciate your patient suggestions, and we feel very strongly that our manuscript is greatly improved as a result of your helpful comments. Hopefully we have addressed all of your concerns.

Round 2
Reviewer 1 Report
Most of the manuscript was improved, but the conclusion was not appropriate.
Line 516: The author concluded that “The environments for resting in the forest can reduce ~.”, but this experiment wasn't done in the forest. The results came from just viewing the scenery of the video of the forest. The situation was not enough to say that the environments as in the forest.
For example, if there were waterfall in reality, the sound would be different and the result would change. The author cannot apply these VR results to the real forest therapy.
So, this conclusion should be revised.
Author Response
International Journal of Environmental Research and Public Health
Manuscript ID: ijerph-561044
Title: The impact of different forest resting environments on stress in Beijing
Many thanks for the comments from you. Below are our responses (in blue) to the comments. We hope that our revised manuscript satisfactorily addresses all issues and that it is now suitable for publication in International Journal of Environmental Research and Public Health. The page and line numbers please refer to our revised manuscript (ijerph-561044-R2.doc).
Most of the manuscript was improved, but the conclusion was not appropriate.
Line 516: The author concluded that “The environments for resting in the forest can reduce ~.”, but this experiment wasn't done in the forest. The results came from just viewing the scenery of the video of the forest. The situation was not enough to say that the environments as in the forest.
For example, if there were waterfall in reality, the sound would be different and the result would change. The author cannot apply these VR results to the real forest therapy.
So, this conclusion should be revised.
Response: We revised it, please see L530.
We are very grateful for your valuable comments, which make our manuscript more rigorous.
Thanks again.
Reviewer 2 Report
The manuscript has been improved and remains a viable study for publication, but I still have questions and concerns.
Updates on 'forest rest' are still unclear (L86-89). Also, as mentioned previously, the multitude of activities specifically meditation means that the natural environment can not reliably be assigned the effects seen here.
I don't see any forest sanitation bases references, nor do I see red ink on line 128.
Regarding objective 3, "more natural" requires operationalization and conceptual definition above.
The names assigned to the 7 environmnets/images are fine, but there is insufficent theoretical background as to why these were selected. For instance, why are there settings with vs without benches/platforms? The platforms perhaps eliciting tension is quite interesting - could you cite something such a relationship exists? And what about benches?
Still, a power analysis should be mentioned in the limitations section if it was not performed. Such an analysis is required by other environmental pscyhology journals for publication. I still do not see multivariate outlier or missing data comments addressed.
I presume "basic data" means baseline.
Fig 10 is still not sufficiently defended. This looks like a path model, rather than a figure that shows what settings were compared to one another.
Author Response
International Journal of Environmental Research and Public Health
Manuscript ID: ijerph-561044
Title: The impact of different forest resting environments on stress in Beijing
Many thanks for the valuable comments. Below are our responses (in blue). We hope that our revised manuscript satisfactorily addresses all issues and that it is now closed for publication in International Journal of Environmental Research and Public Health. The page and line numbers please refer to our revised manuscript (ijerph-561044-R2.doc).
The manuscript has been improved and remains a viable study for publication, but I still have questions and concerns.
Updates on 'forest rest' are still unclear (L86-89). Also, as mentioned previously, the multitude of activities specifically meditation means that the natural environment can not reliably be assigned the effects seen here.
Response: We gave more explanation to forest rest, please see L89-90. For the multitude of activities, we explained more too. Eliminating activities, focusing only on the environment, was to control irrelevant variables and simply explored the impact of the environment. Please see L94-97. But the activities are really important, and we hope that our future research can be specifically targeted at activities.
I don't see any forest sanitation bases references, nor do I see red ink on line 128.
Response: The interpretation of the forest sanitation bases is somewhat difficult for us, and we gave up on this concept. However, regarding the forest sanitation bases, there are many related introductions on the Japanese website, and we have added this reference. Please see L84, 648-649.
Regarding objective 3, "more natural" requires operationalization and conceptual definition above.
Response: We added the operationalization and conceptual definition in L80-82.
The names assigned to the 7 environmnets/images are fine, but there is insufficent theoretical background as to why these were selected. For instance, why are there settings with vs without benches/platforms? The platforms perhaps eliciting tension is quite interesting - could you cite something such a relationship exists? And what about benches?
Response: The theoretical background is a big limitation of this present study, we did not find any suitable theory, even the PSDs. We emphasized this in the discussion section, the limitation part, please see L490-491, L503-504. The name of the environments was according to the actual main landscape elements, which was also not very rigorous. But we didn’t find any better method. Maybe in the future study, with a better theoretical background, all these problems could be improved.
For the platforms and benches, we didn’t find any appropriate reference these days, and we will keep searching. Maybe TSST could explain this to some extent. And it’s very common to see a Chinese person standing on the platform with a flushed face. This result is also very inspiring for us. Perhaps in future experiments, we would specialize in such an environment and combine interviews to explore the laws and causes.
Still, a power analysis should be mentioned in the limitations section if it was not performed. Such an analysis is required by other environmental pscyhology journals for publication. I still do not see multivariate outlier or missing data comments addressed.
Response: We reported this in the results, please see L261-265.
I presume "basic data" means baseline.
Response: We used baseline instead of basic data, please see L229, 266, 271, 276,278, 281.
Fig 10 is still not sufficiently defended. This looks like a path model, rather than a figure that shows what settings were compared to one another.
Response: We deleted Fig 10, just reported the results, please see L363-366.
We are not sure whether we fully understand your comments and response them appropriately. But we are sure that your guidance has greatly improved our manuscript. We are very appreciating for this. Thank you again.
Round 3
Reviewer 2 Report
As mentioned previously, the multitude of activities specifically meditation means that the natural environment can not reliably be assigned the effects seen here.
Response: We gave more explanation to forest rest, please see L89-90. For the multitude of activities, we explained more too. Eliminating activities, focusing only on the environment, was to control irrelevant variables and simply explored the impact of the environment. Please see L94-97. But the activities are really important, and we hope that our future research can be specifically targeted at activities.
NEW COMMENT: The wording of this sentence is awkward and not clear. Also, citations on where people have used these terms would be helpful for people wanting to know the larger body of literature on "forest resting environments."
---
I don't see any forest sanitation bases references, nor do I see red ink on line 128.
Response: The interpretation of the forest sanitation bases is somewhat difficult for us, and we gave up on this concept. However, regarding the forest sanitation bases, there are many related introductions on the Japanese website, and we have added this reference. Please see L84, 648-649.
---
The names assigned to the 7 environmnets/images are fine, but there is insufficent theoretical background as to why these were selected. For instance, why are there settings with vs without benches/platforms? The platforms perhaps eliciting tension is quite interesting - could you cite something such a relationship exists? And what about benches?
Response: The theoretical background is a big limitation of this present study, we did not find any suitable theory, even the PSDs. We emphasized this in the discussion section, the limitation part, please see L490-491, L503-504. The name of the environments was according to the actual main landscape elements, which was also not very rigorous. But we didn’t find any better method. Maybe in the future study, with a better theoretical background, all these problems could be improved.
NEW COMMENT: These sentences are brief and without substance or citations. More reference to applicable environmental psychological theories are warranted, and there are many more than PSD...
---
Author Response
Thank you so much for your comments. We rerevised the manuscript again. Our manuscript has undergone English language editing by MDPI. We hope that the English language and style are fine now. We responded to the new comments, please see the ‘New respond’ below. The page and line numbers please refer to our revised manuscript (ijerph-561044-R3.doc).
As mentioned previously, the multitude of activities specifically meditation means that the natural environment can not reliably be assigned the effects seen here.
Response: We gave more explanation to forest rest, please see L89-90. For the multitude of activities, we explained more too. Eliminating activities, focusing only on the environment, was to control irrelevant variables and simply explored the impact of the environment. Please see L94-97. But the activities are really important, and we hope that our future research can be specifically targeted at activities.
NEW COMMENT: The wording of this sentence is awkward and not clear. Also, citations on where people have used these terms would be helpful for people wanting to know the larger body of literature on "forest resting environments."
New response: We revised the sentence. We added or explained some previous studies for the citations to make it clearer. Please see L97-100.
The names assigned to the 7 environmnets/images are fine, but there is insufficent theoretical background as to why these were selected. For instance, why are there settings with vs without benches/platforms? The platforms perhaps eliciting tension is quite interesting - could you cite something such a relationship exists? And what about benches?
Response: The theoretical background is a big limitation of this present study, we did not find any suitable theory, even the PSDs. We emphasized this in the discussion section, the limitation part, please see L490-491, L503-504. The name of the environments was according to the actual main landscape elements, which was also not very rigorous. But we didn’t find any better method. Maybe in the future study, with a better theoretical background, all these problems could be improved.
NEW COMMENT: These sentences are brief and without substance or citations. More reference to applicable environmental psychological theories are warranted, and there are many more than PSD...
New response: We added more references, please see L76-79.
We learned a lot from your comments, included statistical analysis method and expression, empirically rigorous attitude, extensive substance and citations, etc. Thanks again for all of this.
